# Succinct and Robust Multi-Agent Communication With Temporal Message Control

**Sai Qian Zhang**
Harvard University

**Jieyu Lin**[*]
University of Toronto

**Qi Zhang**[*]
Microsoft

## Abstract

Recent studies have shown that introducing communication between agents can significantly improve overall performance in cooperative Multi-agent reinforcement learning (MARL). However, existing communication schemes often require agents to exchange an excessive number of messages at run-time under a reliable communication channel, which hinders its practicality in many real-world situations. In this paper, we present *Temporal Message Control* (TMC), a simple yet effective approach for achieving succinct and robust communication in MARL. TMC applies a temporal smoothing technique to drastically reduce the amount of information exchanged between agents. Experiments show that TMC can significantly reduce inter-agent communication overhead without impacting accuracy. Furthermore, TMC demonstrates much better robustness against transmission loss than existing approaches in lossy networking environments.

## 1  Introduction

Multi-agent reinforcement learning (MARL) has achieved remarkable success in a variety of challenging problems, including intelligent traffic signal control [36], swarm robotics [15], autonomous driving [28]. At run time, a group of agents interact with each other in a shared environment. Each agent takes an action based on its local observation as well as both direct and indirect interactions with the other agents. This complex interaction model often introduces instability in the training process, which can seriously undermine the overall effectiveness of the model.

In recent years, numerous methods have been proposed to enhance the stability of cooperative MARL. Among these methods, the centralized training and decentralized execution paradigm [21] has gained much attention due to its superior performance. A promising approach to exploit this paradigm is the value function decomposition method [33, 23, 31]. In this method, each agent is given an individual Q-function. A joint Q-function, which is the aggregation of individual Q-functions, is learned during the training phase to capture the cooperative behavior of the agents. During the execution phase, each agent acts independently by selecting its action based on its own Q-function. However, even though value function decomposition has demonstrated outstanding performance in solving simple tasks [33], it does not allow explicit information exchange between agents during execution phrase, which hinders its performance in more complex scenarios. In certain cases, some agents may overfit their strategies to the behaviours of the other agents, causing serious performance degradation [17, 18]. Motivated by the drawbacks due to lack of communication, recent studies [13, 5, 11] have introduced inter-agent communication during the execution phase, which enables agents to better coordinate and react to the environment with their joint experience. However, while an extensive amount of work has concentrated on leveraging communication for better overall performance, little attention has been paid to the reliability of transmission channel and efficiency during the message exchange. Moreover, recent work has shown that the message exchange between agents tends to be excessive and redundant [39]. Furthermore, for many real applications such as autonomous driving and drone

---

[*]Equal contribution, names are ranked alphabetically

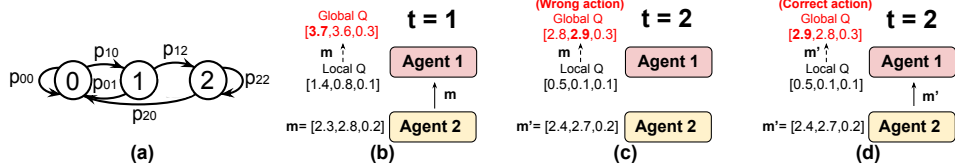

Figure 1: (a) Markov model for wireless loss modeling. (b-d) Impact of small difference on message.

control [30], even though the observations received by the agents change frequently, the useful information presented in the consecutive observations is often quite similar. This further leads to highly time-correlated and redundant information in the output messages. For example, when an autonomous car drives over a spacious mountainous area, its front camera may capture the trees on both sides of the road most of the time. Although the observations are changing continually, it hardly impacts the navigation decision. In fact, these changing observations may cause superfluous and noisy messages to be exchanged, which can reduce the overall system performance. We want the messages to be transmitted only when useful information is captured by the camera (*e.g.*obstacle is detected). Given the limited bandwidth and potentially unreliable communication channels for some of these applications, it is desirable to leverage temporal locality to reduce communication overhead and improve overall system efficiency.

Motivated by this observation, in this paper we present *Temporal Message Control* (TMC), a MARL framework that leverages temporal locality to achieve succinct and robust inter-agent message exchange. Specifically, we introduce regularizers that encourage agents to reduce the number of temporally correlated messages. On the sender side, each agent sends out a new message only when the current message contains relatively new information compared to the previously transmitted message. On the receiver side, each agent stores the most recent messages from the other agents in a buffer, and uses the buffered messages to make action decisions. This simple buffering mechanism also naturally enhances the robustness against transmission loss. We have evaluated TMC on multiple environments including StarCraft Multi-Agent Challenge (SMAC) [25], Predator-Prey [19] and Cooperative Navigation [19]. The results indicate that TMC achieves a $23\%$ higher average winning rates and up to $80\%$ reduction in communication overhead compared to existing schemes. Furthermore, our experiments show that TMC can still maintain a good winning rate under extremely lossy communication channel, where the winning rates of other approaches almost degrade to $0\%$. A video demo is available at [3] to show TMC performance, and the code for this work is provided in [2]. To the best of our knowledge, this is the first study on MARL system that can operate in bandwidth-limited and lossy networking environments.

## 2   Related Work

Given the success of centralized training and decentralized execution scheme [19, 9], the value function decomposition method has been proposed to improve agent learning performance. In Value-Decomposition Network (VDN) [33], the joint Q-value is defined as the sum of the individual agent Q-values. More recently, QMIX [23] use a neural network with non-negative weights to estimate the joint Q-value using both individual Q-values and the global state variables. QTRAN [31] further removes the constraint on non-negative weights in QMIX, and provides a general factorization for the joint Q-value. However, these methods all disallow inter-agent communication.

There has been extensive research [7, 32, 12, 5, 11, 13] on learning effective communication between the agents for performance enhancement of cooperative MARL. However, these methods have not considered quality and efficiency of inter-agent message exchange. Recently, the authors of [39] propose a technique to reduce communication overhead during the execution phase. However, these methods do not consider transmission loss, which limits their applicability in real settings. Kim et. al [14] proposes an efficient training algorithm called *Message-Dropout* (MD). The authors demonstrate that randomly dropping some messages during the training phase can yield a faster convergence speed and make the trained model robust against message loss. However, in practice, the transmission loss pattern is always changing spatially and temporally [34]. Adopting a fixed random loss pattern during training can not generalize to the intricate and dynamic loss pattern during execution. Also, MD incurs high communication overhead, which makes it less practical in real

applications. In contrast, TMC enables the agents to transmit with minimal communication overhead in potentially lossy network environment, while still attaining a good performance.

# 3 Background

**Deep Q-Network**: A standard reinforcement learning problem can be modeled as a Markov Decision Process (MDP). At each timestamp $t \in T$, the agent observes the state $s_t$ and selects an action $a_t$ based on its Q-function $Q(s_t, a_t)$. After carrying out $a_t$, the agent receives a reward $r_t$ and proceeds to the next state $s_{t+1}$. The goal is to maximize the total expected discounted reward $R = \sum_{t=1}^{T} \gamma^t r_t$, where $\gamma \in [0, 1]$ is the discount factor. In Deep Q-network (DQN), a Q-function can be represented by a recurrent neural network [10], namely $Q_\theta(s, h, a) = \sum_{t'=t}^{T} E[r_{t'}|s_{t'} = s, h_{t'-1} = h, a_{t'} = a]$, where $h_{t-1}$ is the hidden states at $t - 1$ and $\theta$ represents the neural network parameters. A replay buffer is adopted to store the transitional tuples $\langle s_t, a_t, s_{t+1}, r_t \rangle$, which are used as training data. The Q-function $Q_\theta(s, h, a)$ can be trained recursively by minimizing the temporal difference (TD) error, which is defined as $L = \sum_{t=1}^{T} (y_t - Q_\theta(s_t, h_{t-1}, a_t))^2$, where $y_t = r_t + \gamma \max_{a'} Q_{\theta^-}(s_{t+1}, h_t, a')$. Here, $\theta^-$ is the parameters of the *target network*. During the training phrase, $\theta^-$ is copied from the $\theta$ periodically, and kept unchanged for several iterations. During execution, the action with maximum Q-value is selected and performed. In this work, we consider a fully cooperative setting where $N$ agents work cooperatively to accomplish a given task in a shared environment. At timestep $t$, each agent $n$ ($1 \le n \le N$) receives a partial observation $o_n^t \in O$, then generates its action $a_n^t$ based on $o_n^t$. After all the agents have completed their actions, they receive a joint reward $r_t$ and proceed to the next timestep $t + 1$. Agents aim at maximizing the joint discounted reward by taking the best actions.

**Value Function Decomposition:** Recent research effort has been made on learning the joint Q-function $Q_{tot}(.)$ for cooperative MARL. In VDN [33], $Q_{tot}(.)$ is represented as the summation of individual Q-functions (*i.e.*, $Q_{tot}(\mathbf{o}_t, \mathbf{h}_{t-1}, \mathbf{a}_t) = \sum_n Q_n(o_n^t, h_n^{t-1}, a_n^t)$), where $\mathbf{o}_t = \{o_n^t\}$, $\mathbf{h}_t = \{h_n^t\}$ and $\mathbf{a}_t = \{a_n^t\}$ are the collections of observations, hidden states and actions from all agents $n \in N$ at timestep $t$. In QMIX [23], the joint Q-function $Q_{tot}(\mathbf{o}_t, \mathbf{h}_{t-1}, \mathbf{a}_t)$ is represented as a nonlinear function of $Q_n(o_n^t, h_n^{t-1}, a_n^t)$ by using a neural network with nonnegative weights.

**Transmission Loss Modeling for Wireless Channel**: Packet loss pattern in wireless network has been studied extensively in the previous works [4, 6, 24]. It has been widely accepted that wireless channel fading and intermediate router traffic congestion are two of the major reasons for wireless packet loss. Moreover, packet loss over wireless channels typically reveals strong temporal correlations and occurs in a bursty fashion [34], which motivates the multi-state Markov model for simulating the loss pattern [35, 16, 40]. Figure 1(a) depicts a three-state Markov model for modeling the wireless packet loss. In this example, state 0 means no loss, state $i$ represents a loss burst of $i$ packet(s), and $p_{ij}$ is the transitional probability from state $i$ to state $j$, which also is the parameter of this model. A loss pattern is generated by performing a Monte Carlo simulation on the Markov model, where state 0 means no packet loss, and all the other states indicate a packet loss.

# 4 Temporal Message Control

In this section, we describe the detailed design of TMC. The main idea of TMC is to improve communication efficiency and robustness by reducing the variation on transmitted messages over time while increasing the confidence of the decisions on action selection.

## 4.1 Agent Network Design

The agent network consists of a *local action generator*, a *message encoder*, a *combining block* and two buffers: a *sent message buffer* and a *received message buffer*. Figure 2(a) shows the network of an agent (agent 1). The received message buffer stores the latest messages received from the teammates. Each stored message is assigned a valid bit that indicates whether the message is expired. The sent message buffer stores the last message sent by agent 1. The local action generator involves a Gated Recurrent Unit (GRU) and a Multilayer Perceptron (MLP). It takes the local observation $o_1^t$ and historical information $h_1^{t-1}$ as inputs and computes the local Q-values $Q_1^{loc}(o_1^t, h_1^{t-1}, a)$ for each possible action $a \in A$. The message encoder contains an MLP. It accepts the intermediate result $c_1^t$ of local action generator and then produces the message $m_1^b = f_{msg}(c_1^t)$ , which is then broadcasted to

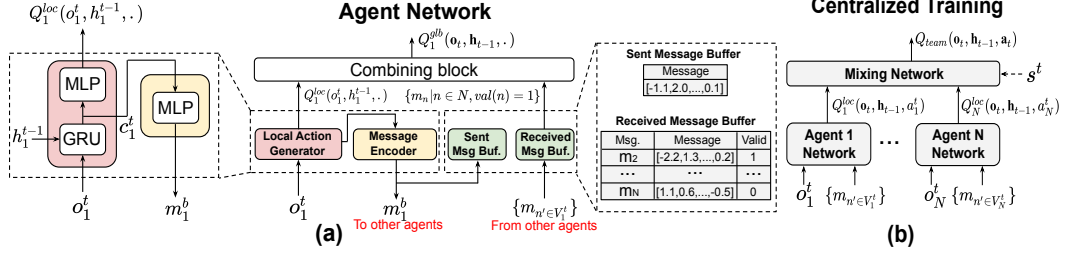

Figure 2: (a) Network for agent 1. $m_n$ denotes the buffered message, $val(n)$ is the valid bit of $m_n$. (b) The centralized training scheme.

the other agents. A copy of the sent message is then saved in the sent message buffer for future usage. On the receiver side, at each timestep $t$, agent 1 receives messages $\{m_{n' \in V_1^t}\} = \{f_{msg}(c_{n' \in V_1^t}^t)\}$ from a subset $V_1^t$ of its teammates. Upon receiving the new messages, agent 1 first updates its received message buffer with the new messages, then selects the messages $m_n$ whose valid bit $val(n)$ is 1. The selected messages $\{m_n | n \in N, val(n) = 1\}$ together with the local Q-values $Q_1^{loc}(o_1^t, h_1^{t-1}, .)$ are passed on to the combining block, which generates the global Q-values $Q_1^{glb}(\mathbf{o}_t, \mathbf{h}_{t-1}, a)$ for each action $a \in A$. The introduction of the messages to the global Q-values generation leads to a better action selection, because the messages contain the information on global observation $\mathbf{o}_t = \{o_n^t | 1 \leq n \leq N\}$ and global history $\mathbf{h}_t = \{h_n^{t-1} | 1 \leq n \leq N\}$ from the other agents. To reduce model complexity and simplify the design, we let the dimension of $Q_1^{loc}(o_1^t, h_1^{t-1}, .)$ and $m_n$ to be identical, so that combining block can simply perform elementwise summation over its inputs, namely $Q_1^{glb}(\mathbf{o}_t, \mathbf{h}_{t-1}, .) = Q_1^{loc}(o_1^t, h_1^{t-1}, .) + \sum_{n \in N, val(n)=1} m_n$. During training, the action is chosen with the $\epsilon$-greedy policy [37]. All the other agents have the identical network architecture as agent 1. To prevent the lazy agent problem [33] and reduce model complexity, all the local action generators and message encoders share their model parameters across agents.

## 4.2  Loss Function Definition

Our key insight for learning succinct message exchange is based on the observations that the message exchanged between agents often show strong temporal locality in most cases. Therefore, we introduce a regularizer to smooth out the messages generated by an agent $n$ within a period of time $w_s$:

$$\mathcal{L}_s^t(\boldsymbol{\theta}) = \sum_{n=1}^{N} \sum_{t'=t_s}^{t-1} \beta_{t-t'} \left[ f_{msg}(c_n^t) - f_{msg}(c_n^{t'}) \right]^2 \tag{1}$$

where $t_s = max(t - w_s, 0)$, and $w_s$ is the smoothing window size. $\boldsymbol{\theta} = \{\theta_{loc}, \theta_{msg}\}$, $\theta_{loc}$ and $\theta_{msg}$ are the model parameters for local action generator and message encoder, respectively, and $\beta_{t-t'}$ is the weight of the square loss term. Equation 1 encourages the messages within the time window to be similar, and therefore agent does not need to transmit the current message if it is highly similar to the messages sent previously. On the other hand, if no message is received from the sender agent, the receiver agent can simply utilize the old message stored in the received message buffer to make its action decisions, as it is very similar to the current message generated by the sender agent.

Unfortunately, using buffered messages for Q-values computation can produce wrong action selections. An example is presented in Figure 1(b-d). Consider a team of two agents, at $t = 1$, agent 2 generates a message $m_2 = [2.3, 2.8, 0.2]$ and sends to agent 1. Agent 1 then uses $m_2$ and $Q_1^{loc}$ to produce the global Q-values $Q_1^{glb}$ (Figure 1(b)). At $t = 2$, agent 2 generates $m_2' = [2.4, 2.7, 0.2]$ which is very close to $m_2$ in Euclidean distance, and therefore $m_2'$ will not be delivered to agent 1. Agent 1 then uses the buffered messages $m_2$ for computing new global Q-values at $t = 2$, which equals $[2.8, 2.9, 0.3]$, and selects the second action (Figure 1(c)), even though the optimal choice should be the first action computed using $m_2'$ (Figure 1(d)). This mistake on action selection is triggered by the proximity between the largest and the second largest Q-values of $Q_1^{glb}$ at $t = 1$, therefore any subtle difference between $m_2$ and $m_2'$ may lead to a wrong selection on the final action. To mitigate this, we encourage the agent to build confidence in its action selection during training, so that it is not susceptible to the small temporal variation of messages. Specifically, we apply another

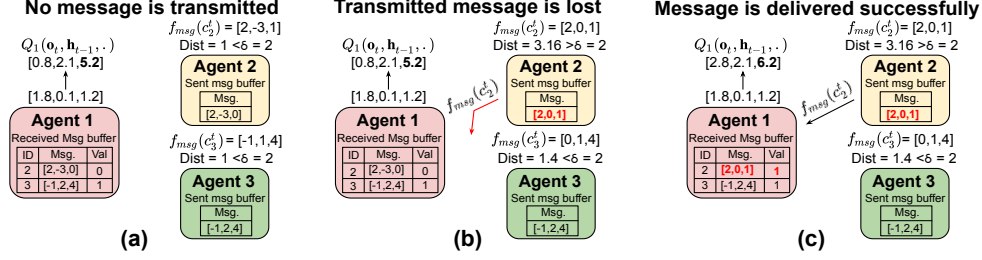

Figure 3: Example on message transmission, communication between agent 2 and 3 is not shown.

regularizer to maximize the action selection confidence, which is defined as the difference between the largest and the second largest elements in the global Q-function:

$$\mathcal{L}_r^t(\boldsymbol{\theta}) = \sum_{n=1}^{N} (e_n^{1,t} - e_n^{2,t})^2 \tag{2}$$

where $e_n^{1,t}$ and $e_n^{2,t}$ are the largest and second largest Q-values in $Q_n^{glb}(\mathbf{o}_t, \mathbf{h}_{t-1}, .)$.

In the value decomposition method, a mixing network is employed to collect global Q-values $Q_n^{glb}(\mathbf{o}_t, \mathbf{h}_{t-1}, a_n^t)$ from the network of each agent $n$, and produces the Q-value $Q_{team}(\mathbf{o}_t, \mathbf{h}_{t-1}, \mathbf{a}_t)$ for the team (Figure 2(b)). Therefore, the loss function $\mathcal{L}(\boldsymbol{\theta})$ during the training phase is defined as:

$$\mathcal{L}(\boldsymbol{\theta}) = \sum_{t=1}^{T} \left[ (y - Q_{team}(\mathbf{o}_t, \mathbf{h}_{t-1}, \mathbf{a}_t; \boldsymbol{\theta}))^2 + \lambda_s \mathcal{L}_s^t(\boldsymbol{\theta}) - \lambda_r \mathcal{L}_r^t(\boldsymbol{\theta}) \right] \tag{3}$$

where $\lambda_s$, $\lambda_r$ are the weights of the regularizers, $y = r_t + \gamma max_{\mathbf{a}_{t+1}} Q_{team}(\mathbf{o}_{t+1}, \mathbf{h}_t, \mathbf{a}_{t+1}; \boldsymbol{\theta}^-)$, and $\boldsymbol{\theta}^-$ is the parameters of the target network which is copied from the $\boldsymbol{\theta}$ periodically. The replay buffer is refreshed periodically by running each agent network and selecting the action under the $\epsilon$-greedy policy.

## 4.3 Communication Protocol Design

The detailed TMC communication protocol is summarized by Algorithm 1. Let $m_n^s$ and $m_{n'}(n' \neq n)$ represent the saved message(s) in the sent message buffer and received message buffer of agent $n$, respectively. Also let $t_n^{last}$ denote the last timestep at which agent $n$ broadcasts messages to the other agents. At timestep $t$, agent $n$ first computes its message $f_{msg}(c_n^t)$ with the current information $c_n^t$. It sends the message only if the Euclidean distance between $f_{msg}(c_n^t)$ and $m_n^s$ is greater than a threshold $\delta$, or a timeout is reached (*i.e.*, $t - t_n^{last} > w_s$, $w_s$ is the smoothing window size). On the receiver side, at timestep $t$, agent $n$ first saves the newly arrived messages (if any) from the agents $n' \in V_n^t$ into the received message buffer, and then checks for message expiration. Specifically, the agent records the time period since each message is last updated, the message is considered as expired if this period is greater than $w_s$ and its valid bit will set to 0. The global Q-value $Q_n^{glb}(\mathbf{o}_t, \mathbf{h}_{t-1}, .)$ is the elementwise summation of the local Q-value $Q_n^{loc}(o_n^t, h_n^{t-1}, .)$ and valid messages $m_{n'}$ in the buffer.

Figure 3 provides an example to demonstrate the protocol behavior under various conditions. Consider a team that consists of 3 agents, each agent has three possible actions to perform. At timestep $t$, agent 1 generates a local Q-value $Q_1^{loc} = [1.8, 0.1, 1.2]$, meanwhile the messages generated by agent 2 and agent 3 are $[2, -3, 1]$ and $[-1, 1, 4]$, respectively (Figure 3(a)). Agent 2 and agent 3 then compute the Euclidean distance between their new messages and the stored messages ($[2, -3, 0]$ and $[-1, 2, 4]$) in the sent message buffers. Assuming timeout is not triggered, since both distances are lower than the threshold $\delta = 2$, no message will be sent by agent 2 and 3. Since agent 1 does not receive any message from the teammates, it uses valid message $[-1, 2, 4]$ in its received message buffer to calculate the global Q-function, $Q_1^{glb} = [0.8, 2.1, 5.2]$. Figure 3(b) shows the scenario when transmission loss occurs. Assume a new message $[2, 0, 1]$ is generated at agent 2, whose Euclidean distance to the buffered message is $3.16 > \delta = 2$. Agent 2 then sends its message to agent 1, and updates its sent message buffer. Assuming the sent message is lost during the transmission, in this case agent 1 will use its current valid buffered message $[-1, 2, 4]$ from agent 3 for computing the

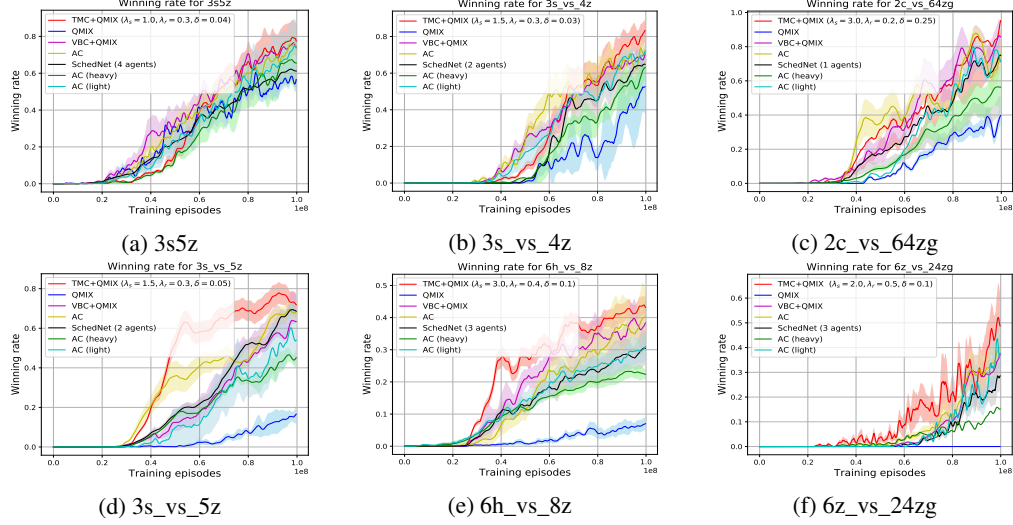

Figure 4: Winning rates for the six scenarios, shaded regions represent the 95% confidence intervals.

global Q-values. Otherwise, if the message from agent 2 is successfully delivered (Figure 3(c)), agent 1 will first update its received message buffer, then uses the latest information to compute its global Q-values $Q_1^{glb}$.

## 4.4  Improvement on Messaging Reliability

---

**Algorithm 1:** Communication protocol at agent $n$

---

**Input**: Last timestep $t_n^{last}$ agent $n$ broadcasts messages, smoothing window size $w_s$. Threshold $\delta$. Set of agents $V_n^t$ which send messages to agent n at t.

**for** $t \in T$ **do**

  // **Sent messages to the rest agents, update the sent message buffer**:

  Generate $f_{msg}(c_n^t)$ based on current information.

  **if** $||f_{msg}(c_n^t) - m_n^s|| \geq \delta \, or \, t - t_n^{last} > w_s$ **then**

    Broadcast message $f_{msg}(c_n^t)$ to other agents.

    Set $m_n^s = f_{msg}(c_n^t)$ and $t_n^{last} = t$.

  // **Update the received message buffer**:

  Receive messages $f_{msg}(c_{n'}^t)$ from agents $n' \in V_n^t$.

  **for** *each message $m_{n'}$ in received msg. buffer* **do**

    **if** $n' \in V_n^t$ **then**

      Set $m_{n'} = f_{msg}(c_{n'}^t)$ and $val(n') = 1$ .

    **if** *$m_{n'}$ has not been updated for $w_s$* **then**

      Set $val(n') = 0$ .

  // **Compute the global Q-values**:

  Compute local Q-values $Q_n^{loc}(o_n^t, h_n^{t-1}, .)$.

  Set $Q_n^{glb}(\mathbf{o}_t, \mathbf{h}_{t-1}, .) = Q_n^{loc}(\mathbf{o}_t, \mathbf{h}_{t-1}, .)$

  **for** *each message $m_{n'}$ in received msg. buffer* **do**

    **if** *val(n') = 1* **then**

      $Q_n^{glb}(\mathbf{o}_t, \mathbf{h}_{t-1}, .) = Q_n^{glb}(\mathbf{o}_t, \mathbf{h}_{t-1}, .) + m_{n'}$

  Select the action with the maximum global Q-value.

---

TMC also enhances the communication robustness against transmission loss, which is a common problem in real world communication networks. Although the reliable communication protocols (*e.g.* TCP) have been developed to ensure the lossless communication, they usually involve a larger communication overhead and latency. For example, TCP allows a reliable communication by retransmitting the lost packets until delivered successfully, where the retransmission timeout (RTO) can be up to several seconds [26]. This is not acceptable for the real-time MARL execution (*e.g.* autonomous driving), where agents need to use the messages instantly for deciding the actions. Additionally, packet retransmission also causes a large communication overhead and delay, which seriously impairs MARL performance. Therefore, we believe that robustness against transmission loss is essential for practical deployment of MARL system. Existing communication schemes (*e.g.*, VBC [39], MAAC [11], etc) have not taken message loss into account when designing their communication protocols. These agents will lose the information in the messages when the transmission loss occurs. In contrast, TMC naturally mitigates the impact of message loss, as each delivered message can be used for at most $w_s$ timesteps, even if the subsequent messages are lost in the future.

# 5 Evaluation on StarCraft Multi-Agent Challenge

In this section, we present the evaluation of TMC on solving the StarCraft Multi-Agent Challenge (SMAC), a popular benchmark recently used by the MARL community [23, 39, 9, 22, 8]. For SMAC, we consider the combat scenario, in which the user's goal is to control the allied units (agents) to destroy all enemy units, while minimizing the total damage taken by allied units. We evaluate TMC in terms of game winning rate, communication overhead, and performance under lossy transmission environment. A video demo is available at [3] to better illustrate the performance of TMC, and the code for this work is provided in [2].

To simulate the lossy transmission, we collect traces on packet loss by conducting real experiments. Specifically, we use an IEEE 802.11ac Access Point (AP) and a pair of Raspberry Pi 3 models [1] as intermediate router, sender and receiver, respectively. We introduce artificial background traffic at the AP to simulate different network conditions. We then make the sender transmit 100000 packets to the receiver, each packet has a payload size of 50 bytes. The experiment is carried out under three different network conditions: *light*, *medium* and *heavy* background traffic. 100 traces are collected under each network condition. We then fit the traces of each network conditions to the Markov model to produce three loss models denoted by $M_{light}$, $M_{medium}$ and $M_{heavy}$. Average loss rate produced by the three models are $1.5\%$, $8.2\%$, $15.6\%$, respectively, which are reasonable for wireless loss [38, 29]. Assume each packet carries one message, the Markov loss model will generate a binary sequence to indicate lost messages during transmission. More details are given in the supplementary materials.

## 5.1 Performance Evaluation

This section summarizes the performance evaluation of TMC on SMAC. We compare TMC with several benchmark algorithms, including: QMIX [23], SchedNet [13] and VBC [39]. We also create an all-to-all communication algorithm (AC), which is derived from TMC by ignoring the two regularizers in equation 3 during training, and remove the limit on message transmission during execution (*i.e.*, $\delta = 0$). For all the methods, we do not include transmission loss during the training phase. Finally, we integrate AC with Message-Dropout [14] by introducing the transmission loss during the training phase of AC. Specifically, we drop the messages between the agents under the loss pattern generated by $M_{light}$ and $M_{heavy}$, and the resulting models are denoted as AC (light) and AC (heavy), respectively. For TMC and VBC, we adopt QMIX as their mixing networks, denoted as TMC+QMIX and VBC+QMIX. For SchedNet, at every timestep we allow roughly half of the agents to send their messages with the $Top(K)$ scheduling policy [13].

We consider a challenging set of cooperative scenarios of SMAC, where the agent group and enemy group consist of (i) 3 Stalkers and 4 Zealots (3s_vs_4z), (ii) 2 Colossus and 64 Zerglings (2c_vs_64zg), (iii) 3 Stalkers and 5 Zealots (3s_vs_5z), (iv) 6 Hydralisks and 8 Zealots (6h_vs_8z), and (v) 6 Zealots and 24 Zerglings (6z_vs_24zg). Finally, we consider a scenario where both agents and enemies consist of 3 Stalkers and 5 Zealots (3s5z). All the algorithms are trained with ten million episodes. We pause the training process and save the model once every 500 training episodes, then run 20 test episodes to measure their winning rates. The transmission loss is not included during training and test phase for all the algorithms except for AC (heavy) and AC (light), where the corresponding loss pattern is involved during both training and test phases. For TMC hyperparameters, we apply $w_s = 6, \beta_1 = 1.5, \beta_2 = 1, \beta_3 = 0.5$ for all 6 scenarios. Other hyperparameters such as $\lambda_r, \lambda_s$ used in equation 3, $\delta$ in Algorithm 1 are shown in the legends of Figure 4. The TMC winning rates are collected by running the proposed protocol described in Algorithm 1.

We train each algorithm 15 times and report the average winning rates and corresponding 95% confidence intervals for the six scenarios in Figure 4. We notice that methods that involve communication (*e.g.*, TMC, VBC, AC, SchdNet) achieve better winning rates than the method without communication (*e.g.*, QMIX), which indicates that communication can indeed benefit performance. Moreover, TMC+QMIX achieves a better winning rate than SchedNet. The reason is that in SchedNet, every timestep only part of the agents are allowed to send messages, therefore some urgent messages can not be delivered in a timely manner. Third, TMC+QMIX obtains a similar performance as AC and VBC in the simple scenarios, and outperforms them in the difficult scenarios (*e.g.*, 3s_vs_5z,6h_vs_8z, 6z_vs_24zg). This is because TMC can efficiently remove the noisy information in the messages with the additional regularizers, which greatly accelerates the convergence of the training process.

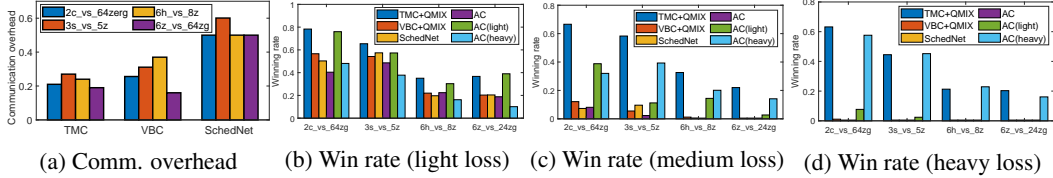

(a) Comm. overhead     (b) Win rate (light loss)   (c) Win rate (medium loss)   (d) Win rate (heavy loss)

Figure 5: (a) Communication overhead comparison. (b-d) Winning rates under different loss patterns.

## 5.2 Communication Overhead

We now evaluate the communication overhead of TMC by comparing it against VBC and SchedNet. QMIX and AC are not included in this evaluation because QMIX agents work independently with no communication, whilst AC agents require all-to-all communication. To quantify the communication overhead, we use a similar approach as used in [39]. Let $x_t$ and $Z$ denote the number of pairs of agents that conduct communications at timestep $t \in T$ and the total number of pairs of agents in the system, respectively. The communication overhead is defined as $\frac{\sum_t x_t}{ZT}$, which is the average number of agent pairs which conducts communication within a single timestep. Assume no loss occurs during message transmission, Figure 5a shows the communication overhead of TMC, VBC and SchedNet for the hard scenarios of SMAC. Compare with VBC and SchedNet, TMC outperforms VBC and SchedNet by $1.3\times$ and $3.7\times$, respectively. This shows that TMC can greatly reduce communication by removing temporal redundant messages.

## 5.3 Performance Under Transmission Loss

Next we evaluate the algorithm performance under communication loss. Specifically, for each method, we run 100 test episodes with their trained models under the three loss patterns generated by $M_{light}$, $M_{medium}$ and $M_{heavy}$, and report the winning rates. For TMC, Algorithm 1 is performed during the execution phase. For all the other methods, if a message is lost, it will simply be set to zero. Figure 5b-5d depict the winning rates of TMC, VBC, SchedNet, AC (light), AC (heavy) under three loss patterns over four maps. We can see there is a severe degradation on the winning rate under all the three loss patterns for VBC, SchedNet and AC. In particular, under the heavy loss, the winning rates of VBC and SchedNet both drop to zero for all the scenarios. Moreover, including loss pattern during training only works if the same loss pattern is utilized during execution. For instance, AC (light) achieves a winning rate of $60\%$ under light loss on 3s_vs_5z, but its winning rate decreases to only $2\%$ under heavy loss. Finally, TMC achieves the best overall performance for all the scenarios under all the three loss patterns, which shows that TMC is robust to dynamic networking loss.

## 5.4 Why is TMC Robust against Transmission Loss?

We investigate the reason behind the robustness of TMC by examining the correlation between the missing messages and the last delivered message. Specifically, we apply the loss pattern generated by $M_{medium}$ to the message transmission during the execution of AC. Note that AC is just a simplified version of TMC without regularizers. For each missing message between a pair of sender and receiver agent, we compute its $l_2$ distance to the last delivered message at the same receiver agent. The blue curve in Figure 6 shows the PDF of this $l_2$ distance on 6h_vs_8z. Then, we train AC by adding the regularizer of equation 1. The new PDF is shown as the red curve in Figure 6. We notice that by adding the smoothing penalty, the lost messages show a higher correlation in $l_2$ distance with the

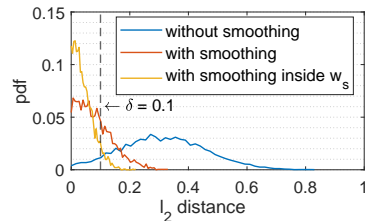

Figure 6: PDF of $l_2$ distance.

last delivered messages. Specifically, $65\%$ of the $l_2$ distances are less than $\delta = 0.1$, where $\delta$ is the threshold for judging similarity in Algorithm 1. Finally, among the lost messages depicted by the red curves, we picked the lost messages which are sent within $w_s$ timesteps from the last delivered messages, the yellow curve shows the new PDF. The correlation becomes even stronger, as $93\%$ of messages has a $l_2$ distance less than 0.1. This shows that even though the following messages are

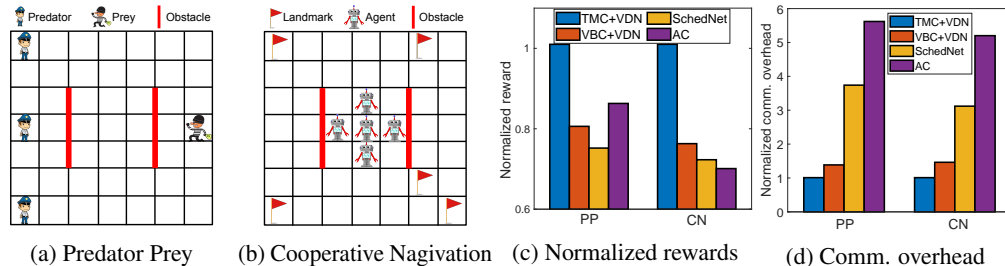

Figure 7: (a) Predator-Prey environment. (b) Cooperative navigation environment. (c) Performance of PP and CN. (d) Communication overhead of PP and CN.

lost, receiver agent can still use the latest delivered message for $w_s$ timesteps, as the lost messages will be similar to it with high probability.

## 6  Evaluation on Predator-Prey and Cooperative Navigation

In this section, we evaluate TMC on Predator-Prey (PP) and Cooperative Navigation (CN), two popular MARL benchmarks which have been widely used by MARL community [19, 12, 5]. Figure 7a and Figure 7b depict the settings for both environments. Both environments are built on a grid world of $7 \times 7$. For PP, three predators try to catch one prey, which moves randomly. The predators win if any of them catch the prey. All predators and prey have the same speed. Each timestep the reward is defined based on the average distances from the predators to the prey, and the predators are also rewarded if any of them catch the prey. We add obstacles (shown in red) in the environment so that the agents can not go across the obstacle. In addition, the communication between a pair of agents will be lost when their line-of-sight is blocked by the obstacle. The predators has a sight range of 3, where they can observe the relative position of their teammates and the target (prey).

In CN, five agents must cooperate to reach five landmarks. Each agent has a $3 \times 3$ horizon, where they can observe the relative positions of the other agents and landmarks within the horizon. For each timestep, the agents are rewarded based on the distance between each agent and the associated landmark, and the agents will be penalized if they collide with each other. Similar to PP, we add the obstacles in the environment so that the agents are not allow to cross the obstacle, and their communication is blocked when their line-of-sight is blocked by the obstacle.

We compare the performance of TMC with VBC, SchedNet and AC on both environments. Because no global state is available during training, we adopt the value decomposition network (VDN) [33] as the mixing network of TMC, VBC and AC. For SchedNet, we allow two agents to broadcast their messages every timestep. We train each method until convergence. For TMC, we apply $w_s = 4$, $\delta = 0.02, \lambda_r = 0.3, \beta_1 = 1.5, \beta_2 = 1$ and $\lambda_s = 1.0$.

Figure 7c depicts the normalized rewards for all the methods on PP and CN. On average, TMC+VDN achieves $1.24\times$ and $1.35\times$ higher reward than the rest algorithms on PP and CN, respectively. This is because TMC can effectively mitigate the impact of transmission loss caused by the line-of-sight blockage. In addition, TMC also achieves the lowest communication overhead, as indicated in Figure 7d. Specifically, TMC+VDN achieves an average of $3.2\times$ and $2.9\times$ reduction on communication overhead on PP and CN compared with the other approaches, respectively.

## 7  Conclusion

In this work, we propose TMC, a cooperative MARL framework that provides significantly lower communication overhead and better robustness against transmission loss. The proposed buffering mechanism efficiently reduces the communication between the agents, also naturally enhances the robustness of the messages transmission against the transmission loss. Evaluations on multiple benchmarks show that TMC achieves a superior performance under different networking environments, while obtaining up to $80\%$ reduction on communication.

## Broader Impact

MARL has been widely applied to many real-world applications, such as autonomous driving [27], game playing [20] and swarm robotics [15]. Recent work [12, 13, 39] have demonstrated that allowing inter-agent communication during execution can greatly enhance the overall performance. However, in practice, the communication channel are usually bandwidth-limited and potentially lossy, which may impact the transmission of the messages and further degrade the overall MARL performance. TMC presents a solution that allows agents to operate under these practical restrictions on communication channel. This is of paramount importance for the adoption of MARL in practice. TMC also paves the way towards enabling MARL applications to work in severe environments caused by natural disasters (*e.g.*, earthquake, tsunami, etc), where the public communication infrastructure is seriously devastated and only limited transmission capability is provided.

However, depending on the intended purpose of the MARL system, TMC can be used either to benefit or hurt social welfare. For instance, terrorists can adopt TMC to control multiple drones for urban terrorism attack. These security concerns can be a major flaw and future research direction for the current version of TMC.

We encourage the machine learning researchers to consider applying the idea of TMC to the other AI fields whose performance heavily depends on quality of the communication channel (*e.g.*, federated learning). We also encourage the researchers from the other related fields (*e.g.*, social science) to investigate the efficient communication pattern for Human-Computer Interaction (HCI) by leveraging the insight of TMC.

## Funding Disclosure

Not available.

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
