[Supplementary Material · TMC_supplementary_materials.pdf]

# Supplementary Materials for Succinct and Robust Multi-Agent Communication With Temporal Message Control

## 1 SMAC Settings and Hyperparameters Descriptions

In this section, we describe the detailed experiment settings of SMAC.

### 1.1 StarCraft micromanagement challenges

We first present the evaluation of TMC on solving the StarCraft Multi-Agent Challenge (SMAC) [7], a popular benchmark which has recently been utilized by MARL community [6, 10, 4, 5, 3]. SMAC focuses on decentralized micromanagement version of Starcraft that involves two armies, one controlled by the user, and the other controlled by the built-in StarCraft II AI. The two armies are placed on the same map and trying to defeat each other. The agent types can be different between the two armies, and the agent types can also be different within the same army. Specifically, we consider the combat scenario, in which the goal of the user is to control the allied units to destroy all enemy units, while minimizing the total damage taken by each individual unit.

We consider six challenging sets of cooperative StarCraft II maps of SMAC, based on their difficulties, they are categorized into three classes: *easy*, hard and *superhard*. Specifically, we consider the following five unsymmetrical scenarios, where the agent group and enemy group each consist of 3 Stalkers and 4 Zealots (3s_vs_4z), 2 Colossus and 64 Zerglings (2c_vs_64zg), 3 Stalkers and 5 Zealots (3s_vs_5z), 6 Hydralisks and 8 Zealots (6h_vs_8z), 6 Zealots and 24 Zerglings (6z_vs_24zg). Finally, we consider a symmetrical scenario where both agent group and enemy group consist of 3 Stalkers and 5 Zealots (3s5z). Among these six scenarios, 3s5z and 3s_vs_4z are categorized as easy, 3s_vs_4z and 2c_vs_64zg are categorized as hard, 6h_vs_8z and 6z_vs_24zg are categorized as superhard.

During execution, each agent are allowed to perform the following set of actions: move[direction], attack[enemy_id], stop and no-op. There are four options for the 'move' operation: south, north, east or west. The number of possible actions for an agent ranges from 10 (3s_vs_4z) to 70 (2c_vs_64zg). In addition, each agent has its own *shooting range* and *sight range*, each agent can only attack the enemies within its shooting range, and can only receive information from the partners or enemies within the sight range. Furthermore, the shooting range is smaller than the sight range so the agent can not attack the opponents without observing them.

Finally, only live agents within the sight range can be observed by the other agents/enemies. At each timestep, the joint reward received by the allied units depends on the total damage on the health levels of the enemy units. In addition, the agents are rewarded 100 extra points after killing each enemy unit, and 200 extra points for killing all the enemies. The allied agents wins only if they kill all the enemies within the time limit. The time limit for different battles are: 150 for 3s5z, 2c_vs_64zg, 3s_vs_4z, and 3s_vs_5z, and 200 for 6h_vs_8z and 6z_vs_24zg.

The observation at each agent contains the following information of all the units within its sight range: relative x,y coordinates, relative distance and agent type. For the mixing network of QMIX, the global state vector $\mathbf{s}_t$ contains the following information:

1. The x,y coordinates of all the units relative to the center of the map at $t$.
2. The actions taken by all the units at $t - 1$.

(a) Changing $\lambda_s$ on 6h_vs_8z

(b) Changing $\lambda_s$ on 6z_vs_64zg

Figure 1: (a) and (b) show the impact of adjusting $\lambda_s$ on communication overhead and the winning rates under different networking conditions on 6h_vs_8z and 6z_vs_24zg. All the communication overheads are normalized. For 6h_vs_8z, $\lambda_r$ and $\sigma$ are fixed at 0.4 and 0.1. For 6z_vs_24zg, $\lambda_r$ and $\sigma$ are fixed at 0.5 and 0.1.

    3. Shield levels, health levels and cooldown levels of all the units at $t$.

Each allied or enemy agent has a sight range of 9 and shooting range of 6 for all types of agents in all the six scenarios. For additional information, please refer to [7].

## 1.2   Hyperparameter for Training

For the network of agent $n$, at timestep $t$, raw observation $o_n^t$ is first passed through a single-layer MLP, which produces an intermediate result with dimension of 64. The GRU then takes this intermediate result, as well as the hidden states $h_n^{t-1}$ from the previous timestep $t-1$, and produces $h_n^t$ and $c_n^t$. Both $h_n^t$ and $c_n^t$ has a dimension of 64. $c_n^t$ is then sent to another a FC layer, which generates the local Q-function $Q_n(o_n^t, h_n^t, .)$. $c_n^t$ is also delivered to the GRU of the message encoder, which takes $c_n^t$ as well as the hidden states $u_n^{t-1}$ and generates a intermediate results with a dimension of 14, then it is passed to a single-layer MLP, which produces the output message.

The discount factor $\gamma$ is set to 0.99. During training, the $\epsilon$ in the action sampling decreases linearly from 1.0 to 0.05 over the first 200000 timesteps and stays at 0.05 for the rest of the training process. The replay buffer stores the most recent 5000 episode. A test run is performed every 200 training episodes to update the replay buffer. The training batch size is set to 32 and the test batch size is set to 8. Finally, a RMSprop optimizer with a learning rate $\eta = 5 \times 10^{-4}$ and $\alpha = 0.99$ is utilized for the training process.

For the hyperparameters used by TMC (i.e., regularization constants $\lambda_s$, $\lambda_r$, threshold on Euclidean distance $\sigma$), we first search for a coarse parameter range based on random trial, experience and message statistics. We then perform a grid search within a smaller hyperparameter space. Best selections are shown in Figure 4 of the paper. For the other hyperparameters, we adopt the default settings of QMIX. For the training, we use a machine with 8 Nvidia 1080Ti GPUs, Intel(R) Xeon(R) CPU E5-2667 v4 at 3.20GHz, 32 cores. The training process takes from 12 to 30 hours.

## 1.3   TMC Performance with Different Hyperparameter

We test the impact of the weight $\lambda_s$ (defined in equation 3) on the TMC performance in terms of communication overhead and winning rate under different networking environments. We show the results on 6h_vs_8z and 6z_vs_24zg, two most difficult scenarios in the SMAC evaluation, and similar results are noticed for the other easy scenarios. We use different $\lambda_s$ for the training, and record the communication overhead and corresponding winning rates under different networking conditions. All the other hyperparameters are kept the same as before. For 6h_vs_8z, as depicted in Figure 1(a), the communication overhead decreases as $\lambda_s$ increases, this is because an increasing $\lambda_s$ will reduce the $l_2$ distance between the consecutive messages, making more message transmission to be erased according to communication protocol. In addition, $\lambda_s = 3.0$ gives the highest winning rate under all the network conditions. Figure 1(b) also presents a similar trend, where $\lambda_s = 2.0$ produces the best overall winning rate on 6z_vs_64zg.

(a) Changing $\sigma$ on 6h_vs_8z        (b) Changing $\sigma$ on 6z_vs_64zg

Figure 2: (a) and (b) show the impact of adjusting the threshold $\sigma$ on communication overhead and the winning rates under different networking conditions on 6h_vs_8z and 6z_vs_24zg. All the communication overheads are normalized by the communication overhead when $\sigma = 0$. For 6h_vs_8z, $\lambda_s$ and $\lambda_r$ are fixed at 3.0 and 0.4. For 6z_vs_24zg, $\lambda_s$ and $\lambda_r$ are fixed at 2.0 and 0.5.

Next, we investigate the effect of the threshold $\sigma$ of Algorithm 1 in the paper by applying different $\sigma$ during the training and fixing all the other hyperparameters. Figure 2 shows the results for 6h_vs_8z and 6z_vs_24zg. The communication overhead decreases as $\sigma$ grows for both scenarios, this is because a greater $\sigma$ will reduce the amount of transmitted messages from the sender agent, as indicated in the communication protocol. For 6h_vs_8z, although $\sigma = 0.05$ produces a slightly higher winning rate than $\sigma = 0.1$, its communication overhead is $2\times$ higher.

## 2 Trace Collection and Loss Modeling

In this section, we describe our trace collection procedure. Our experiment setup involves an IEEE 802.11ac Access Point (AP) and a pair of Raspberry Pi 3 models [1] which are used as intermediate router, sender and receiver, respectively. Each Raspberry Pi device contains a Quad Core 1.2GHz Broadcom BCM2837 64bit CPU, 1GB RAM, and 5GHz IEEE 802.11.b/g/n/ac wireless LAN. We consider the indoor environment (indoor hallway) where obstacles are placed randomly between the sender and receiver. This closely simulates the real environment because the agents usually work under an environment with obstacles. Moreover, even if the environment is spacious and flat, the wireless signal may also be blocked by the other agents in the team.

During experiment, the sender, AP and the receiver are placed in a line, where the AP is located at the midpoint of the sender and receiver. The AP is measured to have a average bandwidth of 8.1Mbps. To simulate the real network environment, we introduce artificial background traffic at AP to consume the bandwidth between the sender and receiver. Therefore we can mock different network conditions by adjusting the amount of background traffic. We then make the sender to transmit 100000 packets to the receiver, each packet has a payload size of 50 bytes. Furthermore, each packet is labelled with a sequence number, so that the receiver can identify the missing packets easily. The experiment is carried out under three different network conditions with light, medium and heavy background traffic. 100 traces are collected under each network condition.

Suggested by [2], we compute the 95th percentiles of the loss run-length across all the traces under each of the three network conditions, and use them as the number of states of the Markov model. We then fit the traces of three network conditions to the Markov model, which produces three Markov models, denoted as $M_{light}$, $M_{medium}$ and $M_{heavy}$, respectively. We then apply these models to simulate the loss between the agents. Average loss rate produced by the three models are $1.5\%$, $8.2\%$, $15.6\%$, respectively, which are reasonable for wireless loss [8, 9]. Assume each packet carries one message, during execution, the Markov model will generate a binary stream, where '1' represents the message is lost and vice versa.

## 3 PDFs on $l_2$ Distance

Figure 3 depicts the pdf curves for the all the scenarios, all the plots indicate a similar tendency as Figure 6 in the paper.

(a) pdf on 3s5z

(b) pdf on 3s_vs_4z

(c) pdf on 2c_vs_64zg

(d) pdf on 3s_vs_5z

(e) pdf on 6h_vs_8z

(f) pdf on 6z_vs_24zg

Figure 3: pdf of $l_2$ distance between the missing messages and the last delivered message.