[Reviews · NeurIPS 2020]

Review 1

Summary and Contributions: This paper presents the Temporal Message Control (TMC) approach for communication in MARL. TMC reduces the amount of information exchanged between agents. It also works well in lossy networking environments.

Strengths: I like the idea of this paper. The idea of not sending messages unless there is new information is very intuitive yet innovative. As the authors say, this naturally makes the communicate more robust in lossy environments. There is no surprise to see that the number of messages is greatly reduced, even though everything else in TMC is quite 'standard.'

Weaknesses: The only weakness that I would like to point out is the evaluation. I agree that there are not standard benchmarks for evaluating, yet the authors should justify why SMAC (and CN) is a not a special case. In particular, the TMC is unlikely to succeed in an MAS that is in chaos. It would also be interesting to know how it performs in an MAS in which agents are first 'misled' to one extreme but eventually mostly return to the 'right' path.

Correctness: Seems so.

Clarity: Yes.

Relation to Prior Work: Yes.

Reproducibility: No

Additional Feedback: I have read the rebuttals from the authors. I am quite satisfied with the rebuttals. The authors could have included in the paper some of the points in the rebuttals, though the inadequacies did not bother me when I reviewed the paper. Therefore, I have revised the overall score to 7.


Review 2

Summary and Contributions: The paper proposes a new algorithm for communication in multi-agent RL based on action value decomposition. The algorithm is shown to be robust against transmission loss in two cooperative learning tasks. UPDATE AFTER REBUTTAL Thank you for addressing my two main criticisms. However, I still maintain my points. (1) The fact remains that messages must be real-vectors and the use of Euclidean distance between messages as well as smoothing over messages in Eq(1) assumes some degree of smoothness between observations. Euclidean distance and smoothness do not work (are not defined) for categorical messages which represent discrete concepts. In your given example, you still use continuous observations with continuous zero-mean noise added, so smoothing should work here too. It could be an interesting direction to try and extend your ideas to also work with categorical messages, to broaden the applicability of your method. (2) The main point is that it's not clear how the use of such a heuristic affects the methods ability to learn correct Q-values, even under ideal conditions such as no message drop. As I note in my review, I found the used heuristics not well justified formally, at least in the paper.

Strengths: Robust RL under unreliable communication settings is an important problem and relevant to NeurIPS as well as other AI/ML sub-communities. The use of illustrations and examples in the algorithm description is useful. The experimental evaluation includes a number of baselines and does ablation studies. It is very nice that the authors fitted the noise model based on real data.

Weaknesses: I think the applicability of this work might be quite limited, since messages must be real-vectors and the use of Euclidean distance between messages as well as smoothing over messages in Eq(1) assumes some degree of smoothness between observations. I believe this is reflected in the environments in which observations are 2D coordinates which are smooth in Euclidean distance. I suspect the algorithm would not work well in other environments that don't satisfy such assumptions. I think this should be discussed in the paper. The method includes some heuristics which in my opinion are not well justified. For example, if no message is received from the sender agent, then the receiver agent simply uses a stored message previously received from the sender. This simple heuristic has clear limitations, and indeed the authors discuss some limitations in the following paragraph. However, the proposed solution is to introduce yet another heuristic in Eq(2) by maximising distance between first and second largest action values. Exactly how does this give robustness against temporal variation of messages, as claimed? And what effect does this have on learning correct Q values?

Correctness: Line 62 is incorrect - early works n MARL already considered communication, such as the 1993 Tan paper. Tan, Ming. "Multi-agent reinforcement learning: Independent vs. cooperative agents." Proceedings of the tenth international conference on machine learning. 1993. Line 83: expecation should be on return, not reward

Clarity: In introduction and related work, the discussion focuses on value decomposition methods in MARL but the reason for this focus is not clear. It only becomes apparent later that your method builds on top of value decomposition methods. This should be clarified early on. Some parts in the paper seems a little redundant, including Sec 3 first paragraph (some redundancy at start/end) and Sec 2 (value decomposition already mentioned in intro). I think these parts can be condensed, and possibly Fig 3 removed, to bring the Cooperative Navigation results back into main paper. In Sec 4, the purpose of some of the method components is not clearly described. It was not clear to me what operations the quantization and combining blocks represent. Line 124: what is the "intermediate result" of the action generator? What does c_1^t denote? Line 126: now c_1^t seems to refer to the message produced by the message encoder? What does the message produced by the message encoder represent? And what is the purpose of quantization? Line 133: how are the global observations and histories obtained? Line 147: I wasn't able to understand this sentence.

Relation to Prior Work: Prior work in MARL communication and the novelty in current paper are discussed.

Reproducibility: Yes

Additional Feedback:


Review 3

Summary and Contributions: The paper proposes a method to reduce the communication overhead in multi-agent reinforcement learning and to deal with transmission loss in lossy network environments. The paper is motivated by the insight that consecutive observations received by an agent are often similar in many applications. So, consecutively communicated messages which are usually the encodings of observations are also similar and redundant. The paper proposes to control message transmissions by setting a threshold to the L2 distance between newly generated message and previously generated messages in a sliding window for each agent. Two regularizers are proposed to smooth out the generated messages and maximize the difference between the largest and the second largest local Q values, respectively. The experiments are performed on StartCraft multi-agent challenge and cooperative navigation (in appendix) where the proposed method is compared with baselines.

Strengths: Using the difference between generated messages to reduce communication redundancy seems new; i.e., when the generated messages are similar, the newly generated message is not to be sent. However, I think whether to send the message more depends on whether it is useful for the receiver. As the paper considers wireless communication where an agent cannot always broadcast messages to all other agents (due to communication range), some agents may not receive the message in previous timesteps. Thus, the message can be particularly important for those agents, although it is similar to previously generated messages. That is to say, it is more reasonable to reduce communication from the perspective of the receiver instead of the sender. The lossy network environment is considered in this paper, which seems new. But I am not sure this is necessary. Why not the message is transmitted by a reliable protocol (like TCP)? Since we already have such protocols, why should we consider the lossy wireless communication underneath?

Weaknesses: I do not understand the purpose of halting training process. None of the methods converge yet. Without the convergence, how to assess the real benefit of the proposed method. The two regularizers serve mainly for communication reduction, and it is not directly correlated with the objective of RL. So, why does TMC+QMIX perform AC, as both use full communication in training). This is not clear, and even counter-intuitive. PP 281-282 mentions “This is because TMC can efficiently remove the noisy information in the messages with the additional regularizers, which greatly accelerates the convergence of the training process.” Yes, this may be true for some scenarios because you halted the training process. What about the converged winning rate? The proposed method seems heavily tuned to compare with baselines. From Figure 4, one can easily see this on \lambda_s, \lambda_r, and \delta for each of the settings. Actually, the method introduces too many hyper-parameters, additionally including \omage, \beta, the bounds of quantization. Only few are investigated by experiments. The ablation study is not thorough. Why not remove the two regularizers individually to investigate the benefit of each? Without such studies, I am not convicted. In the experiments, how many other agents can an agent communicate with? Within a range or all other agents? The paper should consider a scenario where an agent can only communicate with other agents within a limited range to investigate whether the proposed method works in such a scenario as I mention above. As the proposed method is based QMIX, I think the most relevant work is Ref [33] (“Learning nearly decomposable value functions via communication minimization”) that also investigates communication minimization based on QMIX. Why it is not chosen as a baseline? Or could you perform additional experiments to compare with this work?

Correctness: The experiments do not fully support the claims.

Clarity: The paper is easy to follow.

Relation to Prior Work: It is briefly discussed.

Reproducibility: Yes

Additional Feedback: ////////////////////////////////////////// AFTER REBUTTAL The response from the authors addresses many of my comments. Some concerns still remain. First, one important baseline (ref. 33) is missing as I mentioned in the review. But the response does not even mention that. Second, the used heuristic is not fully justified. Therefore, I maintain my score.


Review 4

Summary and Contributions: The authors present Temporal Message Control (TMC), a multi-agent RL (MARL) architecture that aims to enable learning communication protocols in a decentralised systems that have low bandwidth setting requirements. To do this, they extend state of the art centralised-learning-decentralised-execution agents with a buffer system that is trained end to end with the policy and regularised such that the models learn not to overload buffers with similar messages, as well as condensing more useful information in fewer messages. ----- Post-rebuttal edit I would like to thank the authors for the good rebuttal. I am still a little bit worried about the validity of the experimental section wrt. the core contribution of the paper. SMAC is indeed a good general MARL framework, but the pre-built scenarios (and the ones used in the submission) often present quicks in their dynamics and objectives that learning systems can discover and exploit. I think the submission could be strongly improved by evaluating the framework on (a) modified SMAC maps that carefully require the overcoming of information blockage on a per-agent basis, and/or (b) a testbed whose objective requires the agent to develop a good communication protocol -- e.g. consider Hanabi.

Strengths: The problem of learning communication protocols in a multi-agent RL setup is of great interest to the NeurIPS community, as well as being generally a problem that can be quite impactful across various ML and applied fields. Furthermore, the authors choose to tackle problems requirements such as low bandwidth systems that are often blockers for utilising SOTA communication techniques in the real world. The manuscript is well written. The problem setting is explained fairly well, and the contribution, while overall relatively simple, is justified in a reasonable manner. The background section also includes a good coverage of recent methods both wrt. deep MARL methods as well as the more niche "learning to communicate" subcategory. The evaluation is solid, and there's a good amount of ablation qualitative experiments that allow some rationalisation of the effects of the proposed regularisers over the communication buffer.

Weaknesses: Overall, the main weakness of this work steams for the choice of using SMAC for part of the evaluation setup. By relying a lot on a testbed that is mostly used to evaluate MARL systems without communication, it is hard to accurately (and significantly) judge whether TMC is effectively doing what is meant to be doing when compared against other communication learning protocols. This is particularly clear in Figure 4.

Correctness: The overall framework seem functionally correct, and Section 5 does a good job at evaluating how the regularisers affect the resulting policies.

Clarity: The paper is well written. The problem and experimental settings are well described, and the methods, as well as the overall architecture, is mapped both symbolically and in prose. Given that the authors also shared the code, I am fairly confident I could attempt reproducing this work.

Relation to Prior Work: The manuscript does a good a job at describing existing literature in MARL and communication-related work. It also carves a good niche for itself by focusing on placing constraints on the problem that are not commonly considered in other work.

Reproducibility: Yes

Additional Feedback: I'm surprised that the second regulariser was created wrt. the action selection, rather than the message creation. Did you attempt to look into forcing instead the model to send fewer, but more significant messages? Also, why did you choose to use SMAC, instead of strengthening your other experiments (or even going full-in wrt. communication, by for instance employing a benchmark like ParlAI)?

[Author Response · NeurIPS 2020]

We thank the reviewers for their insightful comments. We summarize the questions from each reviewer below and
address them separately. Due to the space limit, we try our best to answer all the major questions from each reviewer.
**(Reviewer 1) Q1: Justify why SMAC is generalizable. TMC may fail for chaotic MAS.**
**A:** SMAC is a challenging benchmark that has been used by recent works to evaluate their MAS communication
schemes [33,37,8,20]. Unlike other environments, SMAC offers partial observability, challenging dynamics, complex
observation and high-dimensional action, which closely simulates the practical scenarios. TMC can perform well under
a chaotic MAS by eliminating noisy information in the observation. Please refer to Q1 of reviewer 2 for more details.
**(Reviewer 2) Q1: TMC may not applicable to the environment where the observation is not smooth.**
**A:** TMC works even if the observation is chaotic. TMC can eliminate the noisy information in the observation, making
message exchange occur only if additional and useful information is presented in the current observation. To show
this, we attach the observation $\mathbf{o}_t$ with a vector of Gaussian noise for 6h_vs_8z scenario, and train TMC with this new
observation $\mathbf{o}'_t$. Figure (a) shows the pdf of $||\mathbf{o}_t - \mathbf{o}_{t-1}||, ||\mathbf{o}'_t - \mathbf{o}'_{t-1}||, ||\mathbf{m}_t - \mathbf{m}_{t-1}||$ and $||\mathbf{m}'_t - \mathbf{m}'_{t-1}||$ across time $t$,
where $\mathbf{m}_t$ and $\mathbf{m}'_t$ are the messages generated with $\mathbf{o}_t$ and $\mathbf{o}'_t$, respectively. We see that the observation becomes very
bumpy after introducing the additional noise, but the generated messages are still smooth (*i.e.*$||\mathbf{m}'_t - \mathbf{m}'_{t-1}||$ is small in
general). Figure (b) depicts the corresponding impact on TMC training process, although the noise slows down the
training initially, TMC can still converge to the same result finally. We will revise the paper to clarify this point.
**(Reviewer 2) Q2: How the regularizer in Eq. 2 gives robustness against message variation and affects Q-values?**
**A:** Let's use an example to clarify. Assume the output of local action generator (local Q-values) at receiver agent has two
elements, denoted as $\mathbf{Q} = [q_1, q_2]$. Assume current message $\mathbf{n} = [n_1, n_2]$ at sender and stored message $\mathbf{m} = [m_1, m_2]$
at receiver are close in $l_2$ distance, so $\mathbf{n}$ will not be transmitted by sender. To infer action, receiver performs elementwise
addition between $\mathbf{Q}$ and $\mathbf{m}$, giving global Q-values $[q_1 + m_1, q_2 + m_2]$. Yet, the more accurate global Q-values should be
$[q_1 + n_1, q_2 + n_2]$. Let $e^1 = q_1 + m_1$ and $e^2 = q_2 + m_2$ denote the largest and second largest Q-values. Eq2 encourages
$e^1 - e^2$ to be large (*i.e.*much greater than 0). Therefore it is highly likely that $e^1 - e^2 > (n_2 - m_2) - (n_1 - m_1)$ (since
$\mathbf{n}$ and $\mathbf{m}$ are close, $(n_2 - m_2) - (n_1 - m_1)$ is close to 0). With some derivations, we get $q_1 + n_1 > q_2 + n_2$. That is,
the regularizer in eq2 allows a better chance of inferring the correct receiver action with the stored message $\mathbf{m}$.
**(Reviewer 2) Q3: Why perform quantization? What is $c_1^t$? How global observations and histories are obtained?**
**A:** We quantize the messages to further reduce communication overhead. $c_1^t$ is the intermediate result of action generator,
which also is the input of message encoder. Global observation and histories are obtained only through the messages.
**(Reviewer 3) Q1: Why not let receiver to initiate the communication? Why not use TCP?**
**A:** Receiver-initiated scheme causes a higher communication overhead and delay, as the receiver needs to first send the
request to senders before getting the messages. This also causes a higher chance for loss, as the communication is failed
if either request or message is lost. In contrast, TMC requires to send the message only, which mitigates above issues.
TCP ensures reliable communication by retransmitting the lost packets until delivered successfully, which is not suitable
for the real-time MARL execution, where agents need to use the messages instantly within the same timestep. Packet
retransmission also causes a large communication overhead and delay, which seriously impairs MARL performance.
**(Reviewer 3) Q2: Why halting training processing? Does TMC+QMIX still outperform AC after convergence?**
We actually pause the training and run 20 test episodes to evaluate the current trained agent network. We then resume
the training process until convergence. As shown in Figure 4 in the paper, TMC+QMIX also wins AC after convergence.
**(Reviewer 3) Q3: Ablation studies should be performed on more hyperparameters (*e.g.*$\lambda_s$, $\lambda_r$,$\sigma$, $w_s$, $\beta$).**
**A:** The ablation studies for $\lambda_s$, $\lambda_r$ and $\sigma$ are given in the appendix. Figure (d) and Table (e) below show the impact of
$w_s$ and $\beta$ on TMC performance for 6h_vs_8z. For $w_s$, although $w_s = 8$ achieves a smaller communication overhead,
$w_s = 6$ gives better winning rates under different loss patterns. For $\beta$, $\beta_1 = 1.5, \beta_2 = 1, \beta_3 = 0.5$ give a slightly better
winning rate, and the communication overhead does not vary too much for the different selection of $\beta_1, \beta_2, \beta_3$.
**(Reviewer 3) Q4: Author should consider the scenario where agents have a limited communication range.**
**A:** We perform a new evaluation by restricting the agents to communicate only within a *communication range*. Figure
(c) below shows the winning rates for 6h_vs_8z by setting communication range to 9, and keeping the agent observation
range to 6. We can see that TMC+QMIX still beats the other methods. This indicates that TMC can always remove the
noisy information in the messages to accelerate the training, even if the communication range is limited.
**(Reviewer 4) Q1: Why do you choose SMAC for evaluation? What does the second regulariser do?**
**A:** SMAC has been used by recent works for testing their communication schemes in MARL [37,33,20,8]. In SMAC,
due to partial observability of each agent, communication plays a key role in learning the advanced techniques (*e.g.*kiting,
besiege), making it a good benchmark for TMC. We will evaluate TMC on more environments (*e.g.*ParlAI) in the future.
During execution, the small difference between the stored message at receiver and true message at sender may cause the
wrong action selection (Fig. 1(b)-(d) in paper). The second regulariser can mitigate this issue (See Q2 of reviewer 2).

(a) Distribution of l2 distance between concecutive observation and message
(b) Performance of TMC on 6h_vs_8z under different observations
(c) Performance with limited communication range on 6h_vs_8z
(d) Impact of Ws under different loss (All other hyperparameters are fixed)
(e) Impact of $\beta$ (assume no message loss, other hyperparameters are fixed)

| $[\beta_1, \beta_2, \beta_3]$ | Win rate | Comm. overhead |
|---|---|---|
| [1.5, 1.0, 0.5] | 55.4% | $1\times$ |
| [1.2, 1.0, 0.7] | 55.0% | $0.98\times$ |
| [1.0, 1.0, 1.0] | 54.5% | $0.97\times$ |
| [0.5, 1.0, 1.5] | 52.7% | $1.05\times$ |

[Meta-Review · NeurIPS 2020]

with scores of (7, 5, 5, 7) this submission turned out to be difficult to adjudicate. All of the reviewers had some positive points to mention in line 2 (Strengths) such as "this is an important topic" and "I like the idea," while R3 argued against the setting. R3 also critiqued the use of heuristics that were not well justified. Correctness was favorable for R1, while R3 said the experiments did not fully support the claims. Clarity and Prior Art coverage were generally good. Reproducibility was also generally thought to be good, except for R1. I should also mention that the reviewers were also pleased to have good empirical results and ablation studies. Given the wide variety of opinions in the reviews in various topics, I am inclined to stick with the numbers provided by the reviewers. This means that the average score will be 6.0. This average score gives the paper a decent chance to be accepted, although acceptance is not at all guaranteed.